# The Differential Response of Intracellular Water Metabolism Derived from Intrinsic Electrophysiological Information in *Morus alba* L. and *Broussonetia papyrifera* (L.) Vent. Subjected to Water Shortage

**Rui Yu** [1], **Yanyou Wu** [2],*, and **Deke Xing** [1]

1   Key Laboratory of Modern Agricultural Equipment and Technology, Ministry of Education, Institute of Agricultural Engineering, Jiangsu University, Zhenjiang 212013, China; 2111316005@stmail.ujs.edu.cn (R.Y.); xingdeke@ujs.edu.cn (D.X.)
2   Research Center for Environmental Bio-Science and Technology, State Key Laboratory of Environmental Geochemistry, Institute of Geochemistry, Chinese Academy of Sciences, Guiyang 550081, China
*   Correspondence: wuyanyou@mail.gyig.ac.cn; Tel.: +86-851-84391746

**Abstract:** Plant electrical signals can quickly respond to the shifting environment. Almost all life activities of plants are dependent on water. The measurement of plant electrophysiological indices provides a more convenient method for studying the intracellular water utilization. In this study, *Morus alba* L. (*Morus alba* or *M. alba*) and *Broussonetia papyrifera* (L.) Vent. (*Broussonetia papyrifera* or *B. papyrifera*) were experimental materials, and the parameters were measured in two habitats (waterfront, well-water and arid slopes, deficient-water). The physiological and electrophysiological responses of leaves to different habitats were analyzed. The theoretically intrinsic relationships between the clamping force and leaf impedance (Z), capacitive reactance (Xc), resistance (R), and inductive reactance (Xl) were revealed as 3-parameter exponential decay and linear models based on bioenergetics, respectively. Leaf intrinsic electrophysiological parameters were successfully obtained by using the above-mentioned relationships and were used to manifest metabolic activity in plants. The intracellular water-holding capacity (IWHC), water use efficiency (IWUE), water-holding time (IWHT), and water transfer rate (WTR) of plant leaves were defined based on the intrinsic electrophysiological parameters and were used to reflect the intracellular water metabolism. The correlation between the physiological and electrophysiological parameters of the two plant species in the two habitats was also analyzed. The results showed that *Morus alba* continuously adapted to the shifting environment, the intracellular water metabolism was insensitive to soil water shortage and was independent from the external physiological state. The intracellular water metabolism in *Broussonetia papyrifera* was very sensitive to soil water shortage, and both intracellular water metabolism and immediate physiological parameters could characterize the response of *Broussonetia papyrifera* growth and development to soil water.

**Keywords:** electrophysiological signals; intracellular water metabolism; bioenergetics; plant physiological information; water shortage response

## 1. Introduction

*Morus alba* L. (*Morus alba* or *M. alba*) and *Broussonetia papyrifera* (L.) Vent. (*Broussonetia papyrifera* or *B. papyrifera*), belonging to the family Moraceae, grow fast, and adapt to adverse environments [1]. The species *M. alba* is the sole food source of the domesticated silkworm and is also an economically important perennial tree [2]. In addition, *M. alba* has edible, medicinal, animal feed, biological materials, ecological protection, and other uses with the developed society and scientific progress [3]. *M. alba* and *B. papyrifera* have great differences in water use efficiency mechanisms and drought resistance. They can be used

as comparison materials to further study the differential response of intracellular water metabolism to water shortage based on electrophysiology.

Photosynthesis in green plants is of great significance to plant growth and development [4]. The photosynthetic rate will decrease under drought stress [5]. Plant leaf water potential is one of the main physiological indicators reflecting plant soil water status. When combined with stomatal conductance and other information, it can determine the plant drought resistance [6,7]. Researchers have reported that the increases of photosynthesis were accompanied by the increased leaf water potential [8,9]. Both photosynthetic indexes and leaf water potential represent the ability of plants to resist drought [10,11]. At the same time, almost all life activities in plants involve charge separation, electron movement, proton and dielectric transport, etc. [12]. The change in water content inevitably leads to changes in membrane permeability and ion concentration inside and outside the cell. Once a plant suffers from environmental stress, the moisture status, ion concentration and membrane permeability of its cells will change immediately, and thereafter change the electrical signal of plant. Therefore, electrophysiological indicators can be used as a theoretical basis for reflecting the water status of plants [13–15]. When the environment changes, plant electrical signals will change correspondingly [16–19]. Therefore, it is of great significance to study the intrinsic electrical parameters of plants and the related environmental factors and physiological responses, in order to evaluate the life phenomena of plant [20,21].

Drought can rapidly trigger plant electrical activity [22]. The ions and ionic groups in plant mesophyll cells are electrolytes, which have been considered concentric spherical capacitors with dual functions of inductance and resistance [23]. The mesophyll cells of plants can record electrical activity caused by external stimuli. Due to the external stimulation, changes in the structure and activity of the plant cell membrane directly affect the physiological process of plants, mainly reflected in changes in the stomatal state, photosynthetic rate and plant electrical signal [24]. In addition to the influence of soil water content on the electrical signals of plant leaves, the clamping force of the two electrodes of a self-made parallel-plate capacitor is also the main factor affecting the electrophysiological parameters [25]. This phenomenon is mainly due to the change in electrolyte concentration of mesophyll cells under different clamping forces. Therefore, the internal relationship between clamping force and electrophysiological parameters can be revealed under different water conditions, providing a fast and real-time method for monitoring the physiological state of plant leaves. At the same time, exploring the correlation between leaf water potential, water content, photosynthesis and electrophysiological information has important practical significance for studying the mechanism of intracellular water utilization in plants.

The water absorbed by roots is transported to the aboveground parts of the plant through vessels in the roots, stems and leaves; only 1–3% of the water is retained in the plant for photosynthesis and other life activities, and almost all of the rest is lost through transpiration [26,27]. To explore the relationship between plants and water, the physiological and molecular mechanisms of the plant response to drought stress are crucial for improving plant productivity and environmental efficiency [28–30]. Therefore, it is necessary to directly and quantitatively monitor the intracellular water status of plant leaves.

In our previous study, we found that the growth and development of *Morus alba* and *Broussonetia papyrifera* respond differently to the environment under short-term drought stress [31]. Can immediate extrinsic parameters and intrinsic electrophysiological parameters characterize the response mechanism of *M. alba* and *B. papyrifera* to water scarcity under prolonged drought conditions? What is the difference between the electrophysiological-based responses of *M. alba* and *B. papyrifera* in their intracellular water metabolism response patterns under water-deficient conditions? Water has a high dielectric constant and is a good condition for various electrolytes in cells to participate in chemical reactions [32]. Inevitably, the water metabolism of plants will not only lead to changes in intracellular and extracellular membrane permeability and ion concentration, but also changes in plant electrical signals [33]. The leaf resistance (R), capacitive reactance (Xc), impedance (Z), and

inductive reactance (Xl) are related to the concentration of ions, ion groups and electric dipoles, and variations in the electrolyte concentration are caused by intracellular water metabolism. This study revealed for the first time the intrinsic mechanism of the clamping force and leaf Z, Xc, R, and Xl of two mulberry species in two habitats and established a physical model on this basis. Subsequently, the intrinsic electrophysiological parameters of the plant leaves were successfully obtained by using these equations. Then the intracellular water-holding capacity (IWHC), water use efficiency (IWUE), water-holding time (IWHT) and water transfer rate (WTR) of the intracellular water metabolism indexes in plant leaves were defined and applied according to the intrinsic electrophysiological parameters. This study aims to reveal the intrinsic mechanistic relationships between leaf Z, Xc, R, and Xl and exogenous stimuli, and provide a novel method for rapid monitoring plant physiological status. At the same time, this study was the first to discuss the correlation between leaf water potential, leaf water content, photosynthesis and electrophysiological information, which was of great significance for revealing plant leaf intracellular water metabolic efficiency.

## 2. Materials and Methods

### 2.1. Experimental Materials

*M. alba* and *B. papyrifera* were grown in two habitats, waterfront and arid slope soil, at Jiangsu University (N 32°11′ and E 119°27′). The average annual temperature, sunshine hours, and precipitation in this test area are 15.4 °C, 2051.7 h, and 1106 mm, respectively. The soil had a total organic matter content of 10.49 g·kg$^{-1}$, a soil field water capacity of 25.5% and a pH value of 7.39. The tested leaves were sampled and measured at 9 a.m. to 11 a.m. on sunny days in June, and the measured temperature was 25.0 ± 2.0 °C.

### 2.2. Measurement of Electrophysiological Parameters of Plant Leaves under Different Clamping Forces

The electrophysiological parameters of plant leaves were measured by an LCR tester (Model 3532-50, Hioki, Nagano, Japan). Taken fully unfolded leaf from the fourth leaf position of each plant. The electrophysiological parameters of leaves were determined according to the method described by Xing et al. [34], the test parameters were further modified and described in our previous studies [25,35] (Figure S1).

### 2.3. Calculation of Intrinsically Electrophysiological Parameters of Plant Leaves

The calculation principle of leaf electrophysiological parameters has been described in our previous study [25,34,36] (Formula S1). According to bioenergetics, the model relationship between plant intrinsic electrophysiological parameters and clamping force was fitted according to the Nernst and Gibbs free energy equation.

### 2.4. Definition of the Intracellular Water Utilization Parameters

According to the model relationship between plant intrinsic electrophysiological parameters and clamping force, intracellular water utilization parameters of leaves were calculated by referring to the previous research method of Zhang et al. [25] (Formula S1).

### 2.5. Determination of Leaf Photosynthetic Parameters, Soil Moisture Content, Leaf Water Content

Net photosynthetic rate ($P_N$, μmol ($CO_2$)m$^{-2}$ s$^{-1}$) and transpiration rate (Tr, mmol m$^{-2}$ s$^{-1}$) of *M. alba* and *B. papyrifera* were measured using li-6400 portable gas exchange measurement system (LI-COR, Lincoln, NE, USA) equipped with a blue/red light source from 9 a.m. to 11 a.m. in the two habitats [37,38]. According to the previous research method of Yu et al. [31], after the determination of the photosynthetic parameters, the soil moisture content and leaf water content of *M. alba* and *B. papyrifera* in the two habitats were measured using the drying method.

## 2.6. Determination of Leaf Water Potential

A water potential system (PSYPRO, Wescor, Inc., Logan, UT, USA) was used to determine the water potential values of *M. alba* and *B. papyrifera* in the two habitats. A hole punch with the corresponding diameter was selected to drill the hole, and it was quickly put into the sample chamber of the C-52 water potential probe. After balancing for 6 min, the measurement began. Three data points were measured each time, and the average value was used as the water potential measurement value of the leaf at this time.

## 2.7. Data Analyses

Data were analyzed using exploratory data analysis by SigmaPlot software (version 10.0, Systat Software Inc., San Jose, CA, USA) and SPSS software (version 21.0, SPSS Inc., Chicago, IL, USA). The statistical analysis included one-way analysis of variance (ANOVA), and significant differences between the means were tested using Tukey's test at 95% confidence. The data are shown as the means $\pm$ SE. Graphs were prepared using Origin Pro. 9.0 (Northampton, MA, USA).

## 3. Results

### 3.1. Soil Moisture Content of M. alba and B. papyrifera in Two Habitats

As illustrated in Table 1, the soil moisture content of *M. alba* and *B. papyrifera* was different in the two habitats, which is, the $T_H$ level was significantly higher than that the $T_L$ level.

**Table 1.** The soil moisture content of *M. alba* and *B. papyrifera* in two habitats.

| Plants | Treatment | $\xi_S$ (H$_2$O) (%) |
|---|---|---|
| *M. alba* | $T_H$ | 21.54 $\pm$ 0.89 a |
| | $T_L$ | 7.90 $\pm$ 0.54 b |
| *B. papyrifera* | $T_H$ | 21.26 $\pm$ 0.64 a |
| | $T_L$ | 7.77 $\pm$ 0.50 b |

Note: values indicate the means $\pm$ SE, $n = 5$. Small letters indicate significant differences at 5% level ($p \leq 0.05$). $T_H$ is waterfront soil. $T_L$ is arid slopes soil.

### 3.2. Fitting Equation Parameters of M. alba and B. papyrifera in Two Habitats

Figure 1 randomly lists the fitting curves and equations of the relationship between leaf Z (Figure 1A), Xc (Figure 1B), R (Figure 1C), Xl (Figure 1D) and champing force (F) in Moraceae. The results show that the relationships of leaf Z, Xc, R, and Xl to the clamping force correlated well. Subsequently, the fitting equation coefficients of both the clamping force and leaf Z, Xc, R, and Xl in *M. alba* and *B. papyrifera* grown in two habitats were calculated separately (Table 2). The correlation coefficients ($R^2$) of the fitting equations of Z-F, Xc-F, R-F, and Xl-F for sixteen leaves of *B. papyrifera* in two habitats were 0.9709–0.9938, 0.9769–0.9949, 0.9620–0.9940, and 0.9770–0.9943, and those in *M. alba* in two habitats were 0.9733–0.9937, 0.9673–0.9931, 0.9764–0.9960, and 0.9676–0.9933, respectively. Moreover, all *p* values were less than 0.0001. This result shows that the fitting equations of Z-F, Xc-F, R-F, and Xl-F had good correlations, indicating the real existence of intrinsic mechanism relations between F and leaf Z, Xc, R, and Xl.

### 3.3. Electrophysiological Information of M. alba and B. papyrifera in Two Habitats

As shown in Table 3, the intrinsic electrophysiological parameters of *M. alba* and *B. papyrifera* in the two habitats are depicted, and the leaf intrinsic impedance (IZ), intrinsic resistance (IR), intrinsic inductive reactance (IXl), capacitive reactance (IXc), and intrinsic capacitance (IC) of *M. alba* at the $T_H$ and $T_L$ levels were not significantly different. Moreover, the leaf IZ, IR, IXl, and IXc of *B. papyrifera* at the $T_H$ levels were significantly lower than those at the $T_L$ levels, and the leaf IC of *B. papyrifera* at the $T_H$ levels was significantly higher than that at the $T_L$ levels. The IZ, IR, IXl, and IXc values of *B. papyrifera* at the $T_L$ level were significantly increased by 325%, 335%, 331%, and 313%, respectively, compared to the $T_H$

levels. However, the IC values of *B. papyrifera* at $T_L$ levels were significantly decreased by 77% compared to the $T_H$ levels.

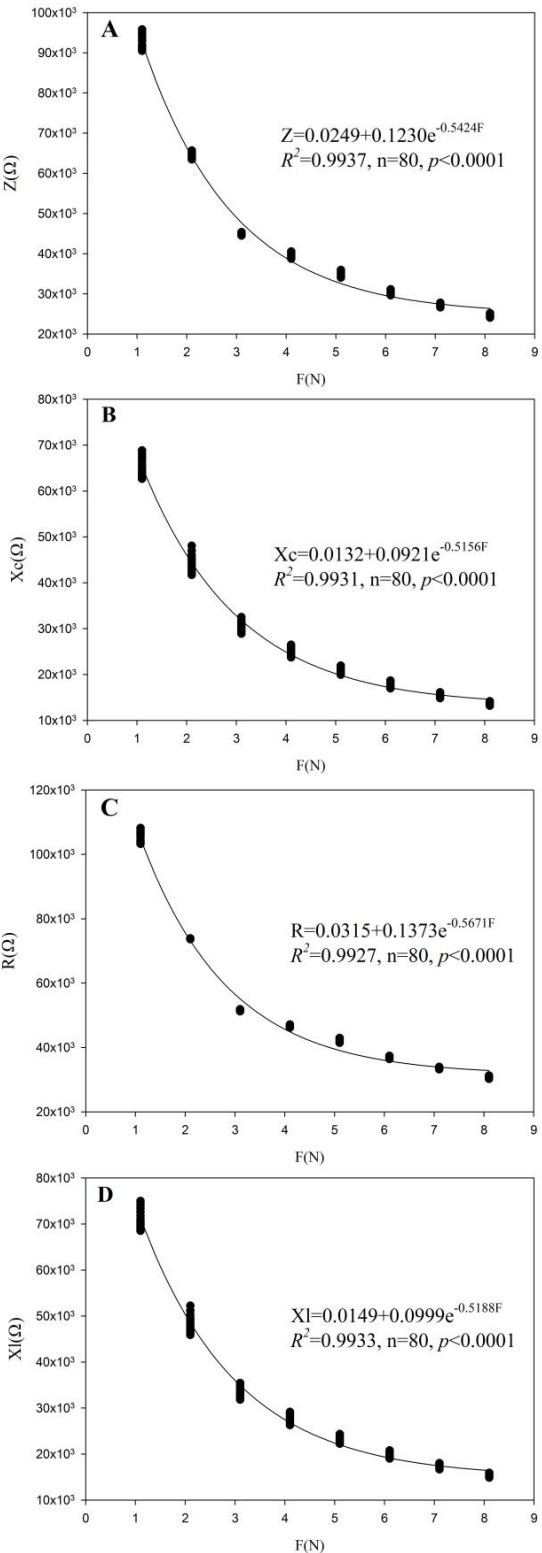

**Figure 1.** Fitting curves and equations of the relationships among leaf Z (**A**), Xc (**B**), R (**C**), Xl (**D**), and champing force (F) in Moraceae.

**Table 2.** Fitting equation parameters of *M. alba* and *B. papyrifera*.

| Plants | Treatment | Z-F | Xc-F | R-F | Xl-F |
|--------|-----------|-----|------|-----|------|
| | | $y_0/k_1/b_1$ R$^2$/$p$< | $p_0/k_2/b_2$ R$^2$/$p$< | $g_0/k_3/b_3$ R$^2$/$p$< | $q_0/k_4/b_4$ R$^2$/$p$< |
| *M. alba* | T$_H$-1 | 0.0249/0.1230/0.5424 0.9937/0.0001 | 0.0132/0.0921/0.5156 0.9931/0.0001 | 0.0315/0.1373/0.5671 0.9927/0.0001 | 0.0149/0.0999/0.5188 0.9933/0.0001 |
| | T$_H$-2 | 0.0288/0.2364/0.5862 0.9925/0.0001 | 0.0173/0.2489/0.7858 0.9674/0.0001 | 0.0365/0.2418/0.5473 0.9960/0.0001 | 0.0194/0.2611/0.7665 0.9700/0.0001 |
| | T$_H$-3 | 0.0178/0.1831/0.5822 0.9779/0.0001 | 0.0072/0.0874/0.4052 0.9795/0.0001 | 0.0230/0.2495/0.6632 0.9764/0.0001 | 0.0088/0.0998/0.4278 0.9792/0.0001 |
| | T$_H$-4 | 0.0255/0.2368/0.5748 0.9878/0.0001 | 0.0177/0.2925/0.6980 0.9845/0.0001 | 0.0296/0.2419/0.5606 0.9879/0.0001 | 0.0195/0.3059/0.6904 0.9847/0.0001 |
| | T$_L$-1 | 0.0337/0.1395/0.5515 0.9922/0.0001 | 0.0236/0.1842/0.7192 0.9832/0.0001 | 0.0393/0.1409/0.5402 0.9914/0.0001 | 0.0258/0.1915/0.7078 0.9842/0.0001 |
| | T$_L$-2 | 0.0371/0.0561/0.3261 0.9918/0.0001 | 0.0290/0.1020/0.3708 0.9908/0.0001 | 0.0413/0.0547/0.3380 0.9910/0.0001 | 0.0314/0.1052/0.3696 0.9909/0.0001 |
| | T$_L$-3 | 0.0537/0.2467/0.9774 0.9733/0.0001 | 0.0337/0.1832/0.6666 0.9673/0.0001 | 0.0622/0.2995/1.1238 0.9783/0.0001 | 0.0373/0.1972/0.6847 0.9676/0.0001 |
| | T$_L$-4 | 0.0277/0.1153/0.8068 0.9792/0.0001 | 0.0167/0.1170/0.7476 0.9797/0.0001 | 0.0332/0.1279/0.8939 0.9789/0.0001 | 0.0186/0.1240/0.7530 0.9797/0.0001 |
| *B. papyrifera* | T$_H$-1 | 0.0298/0.2235/0.6178 0.9889/0.0001 | 0.0145/0.1875/0.6226 0.9844/0.0001 | 0.0403/0.2442/0.6525 0.9889/0.0001 | 0.0167/0.2014/0.6237 0.9850/0.0001 |
| | T$_H$-2 | 0.0252/0.1182/0.4428 0.9933/0.0001 | 0.0129/0.1109/0.5128 0.9899/0.0001 | 0.0343/0.1210/0.4478 0.9910/0.0001 | 0.0148/0.1179/0.5084 0.9905/0.0001 |
| | T$_H$-3 | 0.0277/0.2136/0.5703 0.9914/0.0001 | 0.0138/0.1335/0.5496 0.9853/0.0001 | 0.0369/0.2539/0.5912 0.9932/0.0001 | 0.0158/0.1477/0.5528 0.9863/0.0001 |
| | T$_H$-4 | 0.0349/0.1283/0.3730 0.9938/0.0001 | 0.0194/0.0936/0.4646 0.9887/0.0001 | 0.0438/0.1399/0.3266 0.9940/0.0001 | 0.0219/0.1013/0.4534 0.9894/0.0001 |
| | T$_L$-1 | 0.0980/0.5444/0.5290 0.9793/0.0001 | 0.0513/0.3909/0.4726 0.9949/0.0001 | 0.1225/0.6150/0.5545 0.9844/0.0001 | 0.0582/0.4256/0.4785 0.9943/0.0001 |
| | T$_L$-2 | 0.1448/0.6665/0.6820 0.9840/0.0001 | 0.0880/0.5156/0.5623 0.9858/0.0001 | 0.1700/0.7525/0.7302 0.9804/0.0001 | 0.0975/0.5559/0.5698 0.9851/0.0001 |
| | T$_L$-3 | 0.1104/0.7424/0.4926 0.9807/0.0001 | 0.0794/0.6497/0.5687 0.9861/0.0001 | 0.1461/0.9212/0.6337 0.9944/0.0001 | 0.0937/0.7578/0.6509 0.9844/0.0001 |
| | T$_L$-4 | 0.1176/0.9636/0.8204 0.9709/0.0001 | 0.0635/0.6419/0.7601 0.9769/0.0001 | 0.1480/1.1231/0.8556 0.9620/0.0001 | 0.0717/0.7038/0.7656 0.9770/0.0001 |

Note: Z: impedance, Xc: capacitive reactance, R: resistance, Xl: inductive reactance, F: clamping force. T$_H$ is waterfront soil. T$_L$ is arid slopes soil.

**Table 3.** The electrophysiological parameters of *M. alba* and *B. papyrifera* in the two habitats.

| Plants | Treatment | IZ (MΩ) | IR (MΩ) | IXl (MΩ) | IXc (MΩ) | IC (pF) |
|--------|-----------|---------|---------|----------|----------|---------|
| *M. alba* | T$_H$ | 0.22 ± 0.03 b | 0.25 ± 0.03 b | 0.21 ± 0.06 b | 0.19 ± 0.06 b | 358.55 ± 44.96 a |
| | T$_L$ | 0.18 ± 0.04 b | 0.20 ± 0.06 b | 0.18 ± 0.03 b | 0.17 ± 0.02 b | 325.33 ± 43.61 a |
| *B. papyrifera* | T$_H$ | 0.20 ± 0.03 b | 0.23 ± 0.03 b | 0.16 ± 0.02 b | 0.15 ± 0.02 b | 380.16 ± 45.24 a |
| | T$_L$ | 0.85 ± 0.09 a | 1.00 ± 0.11 a | 0.69 ± 0.08 a | 0.62 ± 0.07 a | 88.96 ± 10.86 b |

Note: values indicate the means ± SE, $n$ = 5. Small letters indicate significant differences at 5% level ($p \leq 0.05$). Different lowercase letters in the same column indicate significant differences in measurement factors. IZ: intrinsic impedance, IXc: intrinsic capacitive reactance, IR: intrinsic resistance, IXl: intrinsic inductive reactance, IC: intrinsic capacitance. T$_H$ is waterfront soil. T$_L$ is arid slopes soil.

*3.4. Intracellular Water Utilization of M. alba and B. papyrifera in Two Habitats*

The water use parameters of *M. alba* and *B. papyrifera* in the two habitats were obtained by using the corresponding parameters of the fitting equations. The results showed that there were significant differences in leaf specific effective thickness (d), IWUE, IWHC, IWHT, WTR, and water content of *M. alba* and *B. papyrifera* between the two habitats. As shown in Table 4, the leaf d of *M. alba* at the T$_H$ levels was significantly higher than that at the T$_L$ levels, the leaf d values of *M. alba* at the T$_L$ levels were significantly decreased by 49% compared to the T$_H$ levels, but the leaf IWUE, IWHT, and WTR of *M. alba* in the two habitats were not significantly different. However, the leaf d and WTR of *B. papyrifera* at the T$_H$ levels were significantly higher than those at the T$_L$ levels, and the d and WTR values of *B. papyrifera* at the T$_L$ levels were significantly decreased by 78% and 89%, respectively,

compared to the $T_H$ levels, but the leaf IWUE and IWHT of *B. papyrifera* in the two habitats were not significantly different.

**Table 4.** Water use parameters of *M. alba* and *B. papyrifera* in the two habitats.

| Plants | Treatment | d | IWUE | IWHT | WTR |
|---|---|---|---|---|---|
| *M. alba* | $T_H$ | 547.54 ± 58.45 a | 0.12 ± 0.04 a | 71.18 ± 5.14 a | 93.18 ± 25.03 a |
| | $T_L$ | 279.29 ± 61.00 b | 0.05 ± 0.01 a | 53.03 ± 7.88 a | 124.90 ± 35.21 a |
| *B. papyrifera* | $T_H$ | 468.86 ± 55.19 a | 0.07 ± 0.02 a | 72.88 ± 5.65 a | 104.93 ± 19.79 a |
| | $T_L$ | 102.62 ± 12.17 c | 0.13 ± 0.02 a | 72.94 ± 4.17 a | 11.66 ± 1.94 b |

Note: values indicate the means ± SE, $n = 5$. Small letters indicate significant differences at 5% level ($p \leq 0.05$). Different lowercase letters in the same column indicate significant differences in measurement factors. d: Specific effective thickness, IWHT: intracellular water-holding time, IWUE: intracellular water use efficiency, WTR: dynamic water transfer rate. $T_H$ is waterfront soil. $T_L$ is arid slopes soil.

As shown in Figure 2, the IWHC and leaf water content of *M. alba* and *B. papyrifera* in the two habitats were displayed and basically consistent. The leaf IWHC and leaf water content of *M. alba* and *B. papyrifera* at the $T_H$ levels were significantly higher than those at the $T_L$ levels. The leaf IWHC values of *M. alba* and *B. papyrifera* at the $T_L$ level were significantly decreased by 19% and 89%, respectively, compared to those at the $T_H$ level. The leaf water content values of *M. alba* and *B. papyrifera* at the $T_L$ level were significantly decreased by 18% and 25%, respectively, compared to those at the $T_H$ level.

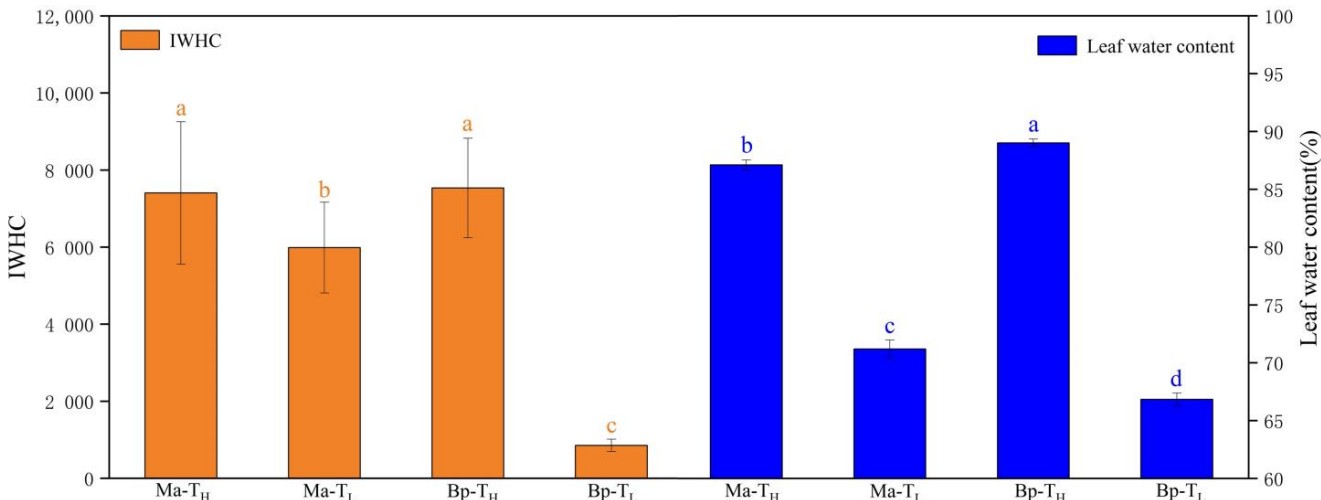

**Figure 2.** IWHC and leaf water content of *M. alba* and *B. papyrifera* in the two habitats. Note: values indicate the means ± SE, $n = 5$. Small letters indicate significant differences at 5% level ($p \leq 0.05$). IWHC: intracellular water-holding capacity, Bp-$T_H$: *B. papyrifera* at the $T_H$ level, Bp-$T_L$: *B. papyrifera* at the $T_L$ level, Ma-$T_H$: *M. alba* at the $T_H$ level, Ma-$T_L$: *M. alba* at the $T_L$ level.

*3.5. Photosynthetic Parameters of M. alba and B. papyrifera in Two Habitats*

The net photosynthetic rate (Figure 3A) and transpiration rate (Figure 3B) of the leaves of *M. alba* and *B. papyrifera* in the two habitats are shown in Figure 4. The net photosynthetic rate and transpiration rate of the leaves of *M. alba* and *B. papyrifera* at the $T_H$ levels were significantly higher than those at the $T_L$ levels. The leaf $P_N$ values of *M. alba* and *B. papyrifera* at the $T_L$ level were significantly decreased by 23% and 26%, respectively, compared to those at the $T_H$ level. The leaf Tr values of *M. alba* and *B. papyrifera* at the $T_L$ level were significantly decreased by 27% and 33%, respectively, compared to those at the $T_H$ level.

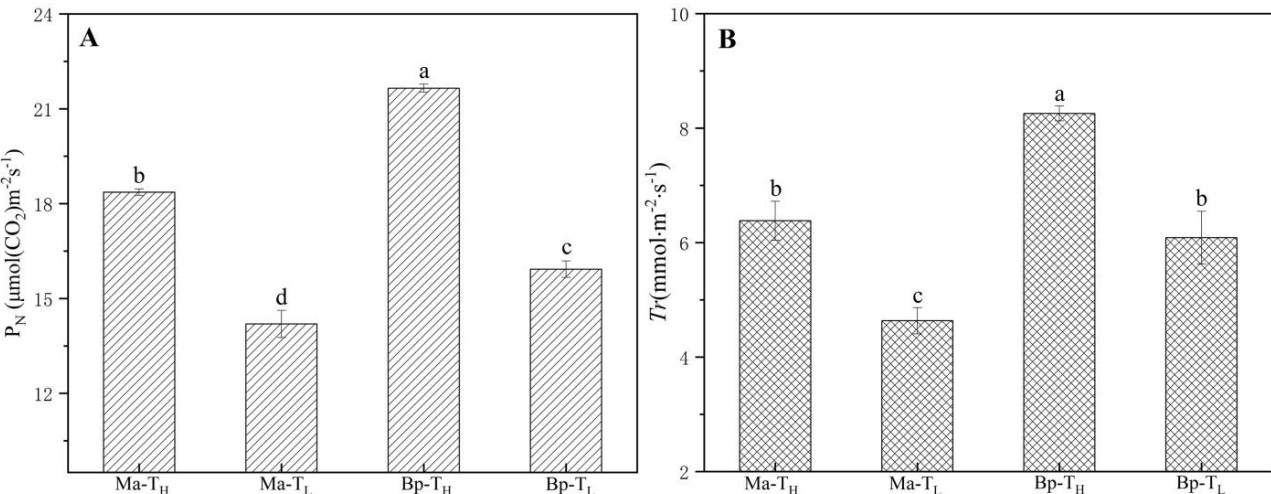

**Figure 3.** The net photosynthetic rate ($P_N$) (**A**) and transpiration rate (*Tr*) (**B**) of the leaves of *M. alba* and *B. papyrifera* in the two habitats. Note: Values indicate the means $\pm$ SE, $n = 5$. Small letters indicate significant differences at 5% level ($p \leq 0.05$). Bp-$T_H$: *B. papyrifera* at the $T_H$ level, Bp-$T_L$: *B. papyrifera* at the $T_L$ level, Ma-$T_H$: *M. alba* at the $T_H$ level, Ma-$T_L$: *M. alba* at the $T_L$ level.

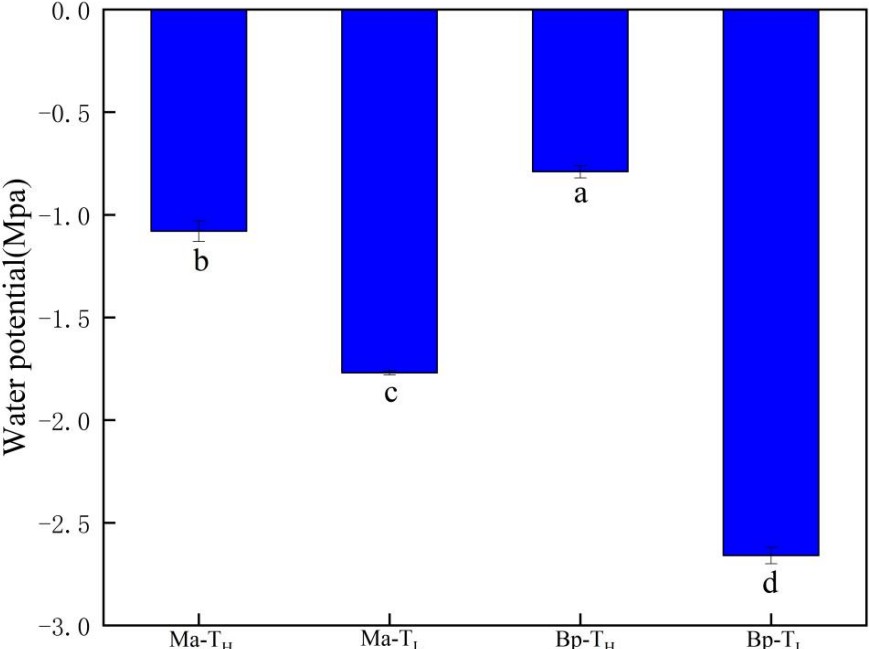

**Figure 4.** The leaf water potential of *M. alba* and *B. papyrifera* in the two habitats. Note: values indicate the means $\pm$ SE, $n = 5$. Small letters indicate significant differences at 5% level ($p \leq 0.05$). Bp-$T_H$: *B. papyrifera* at the $T_H$ level, Bp-$T_L$: *B. papyrifera* at the $T_L$ level, Ma-$T_H$: *M. alba* at the $T_H$ level, Ma-$T_L$: *M. alba* at the $T_L$ level.

*3.6. Leaf Water Potential of M. alba and B. papyrifera in Two Habitats*

The leaf water potential of *M. alba* and *B. papyrifera* in the two habitats is presented in Figure 4. The leaf water potential of the leaves of *M. alba* and *B. papyrifera* at the $T_H$ levels was obviously higher than that at the $T_L$ levels, with significant differences observed between the two habitats. The leaf water potential values of *M. alba* and *B. papyrifera* at the $T_L$ level were significantly decreased by 39% and 70%, respectively, compared to those at the $T_H$ level.

### 3.7. Correlation of Different Physiological Information and Intracellular Water Utilization Parameters

The Pearson correlation coefficients for the relationships between the different physiological information and intracellular water utilization parameters of *M. alba* and *B. papyrifera* are shown in Tables 5 and 6. In *M. alba* (Table 5), IZ showed a strong positive correlation with IR, strong negative correlation with WTR, and positive correlation with IXl and IXc. IR showed strong negative correlation with WTR. IXl was found to exhibit strong positive correlation with IXc and IWUE, strong negative correlation with IC and IWHC, and negative correlation with WTR. IXc showed strong positive correlation with IWUE, strong negative correlation with IC and IWHC, and negative correlation with WTR. IC was found to exhibit strong positive correlation with IWHC and negative correlation and IWUE. D showed positive correlation with $\Psi$ and $\xi_L$. IWUE was negative correlation with WTR. $\Psi$ showed strong positive correlation with $\xi_L$ and $P_N$. $\xi_L$ showed strong positive correlation with $P_N$. $P_N$ exhibited strong positive correlation with Tr.

**Table 5.** Pearson correlation coefficients among different physiological parameters of *M. alba*.

| | IR | IXl | IXc | IC | d | IWHC | IWUE | IWHT | WTR | Ψ | ξ_L | P_N | Tr |
|---|---|---|---|---|---|---|---|---|---|---|---|---|---|
| IZ | 0.961 ** | 0.741 * | 0.713 * | −0.622 | 0.131 | −0.583 | 0.624 | 0.184 | −0.940 ** | 0.289 | 0.260 | 0.261 | 0.123 |
| IR | | 0.537 | 0.501 | −0.404 | 0.215 | −0.364 | 0.432 | 0.426 | −0.857 ** | 0.277 | 0.245 | 0.247 | 0.069 |
| IXl | | | 0.999 ** | −0.951 ** | −0.151 | −0.927 ** | 0.869 ** | −0.477 | −0.806 * | 0.176 | 0.152 | 0.157 | 0.082 |
| IXc | | | | −0.957 ** | −0.169 | −0.935 ** | 0.868 ** | −0.512 | −0.785 * | 0.163 | 0.140 | 0.146 | 0.080 |
| IC | | | | | 0.397 | 0.998 ** | −0.714 * | 0.642 | 0.705 | 0.119 | 0.131 | 0.073 | 0.026 |
| d | | | | | | 0.449 | 0.295 | 0.649 | −0.189 | 0.761 * | 0.799 * | 0.530 | 0.319 |
| IWHC | | | | | | | −0.671 | 0.676 | 0.669 | 0.183 | 0.193 | 0.129 | 0.057 |
| IWUE | | | | | | | | −0.302 | −0.720 * | 0.549 | 0.550 | 0.406 | 0.259 |
| IWHT | | | | | | | | | −0.061 | 0.373 | 0.372 | 0.307 | 0.115 |
| WTR | | | | | | | | | | −0.271 | −0.252 | −0.213 | −0.122 |
| Ψ | | | | | | | | | | | 0.977 ** | 0.839 ** | 0.573 |
| ξ_L | | | | | | | | | | | | 0.883 ** | 0.667 |
| P_N | | | | | | | | | | | | | 0.861 ** |

Note: ** correlation is significant at the 0.01 level (two-tailed). * Correlation is significant at the 0.05 level (two-tailed).

**Table 6.** Pearson correlation coefficients among different physiological parameters of *B. papyrifera*.

| | IR | IXl | IXc | IC | d | IWHC | IWUE | IWHT | WTR | Ψ | ξ_L | P_N | Tr |
|---|---|---|---|---|---|---|---|---|---|---|---|---|---|
| IZ | 0.998 ** | 0.970 ** | 0.979 ** | −0.930 ** | −0.858 ** | −0.901 ** | 0.783 * | 0.082 | −0.893 ** | −0.938 ** | −0.931 ** | −0.979 ** | −0.885 ** |
| IR | | 0.979 ** | 0.985 ** | −0.927 ** | −0.854 ** | −0.897 ** | 0.792 * | 0.066 | −0.891 ** | −0.935 ** | −0.931 ** | −0.977 ** | −0.894 ** |
| IXl | | | 0.998 ** | −0.931 ** | −0.860 ** | −0.899 ** | 0.756 * | −0.098 | −0.880 ** | −0.941 ** | −0.937 ** | −0.951 ** | −0.917 ** |
| IXc | | | | −0.939 ** | −0.866 ** | −0.908 ** | 0.758 * | −0.077 | −0.889 ** | −0.948 ** | −0.942 ** | −0.960 ** | −0.917 ** |
| IC | | | | | 0.786 * | 0.996 ** | −0.769 * | 0.038 | 0.977 ** | 0.947 ** | 0.921 ** | 0.866 ** | 0.782 * |
| d | | | | | | 0.738 * | −0.400 | −0.033 | 0.748 * | 0.918 ** | 0.931 ** | 0.922 ** | 0.694 |
| IWHC | | | | | | | −0.773 * | 0.030 | 0.981 ** | 0.922 ** | 0.891 ** | 0.823 * | 0.745 * |
| IWUE | | | | | | | | 0.004 | −0.746 * | −0.617 | −0.586 | −0.657 | −0.734 * |
| IWHT | | | | | | | | | −0.126 | 0.032 | 0.015 | −0.111 | 0.169 |
| WTR | | | | | | | | | | 0.901 ** | 0.875 ** | 0.826 * | 0.710 * |
| Ψ | | | | | | | | | | | 0.996 ** | 0.933 ** | 0.765 * |
| ξ_L | | | | | | | | | | | | 0.939 ** | 0.759 * |
| P_N | | | | | | | | | | | | | 0.869 ** |

Note: ** correlation is significant at the 0.01 level (two-tailed). * Correlation is significant at the 0.05 level (two-tailed).

In *B. papyrifera* (Table 6), IZ showed strong positive correlation with IR, IXl and IXc, was highly negatively correlated with IC, d, IWHC, WTR, $\Psi$, $\xi_L$, $P_N$, and Tr, and positively correlated with IWUE. IR showed strong positive correlation with IXl and IXc, was strongly negatively correlated with IC, d, IWHC, WTR, $\Psi$, $\xi_L$, $P_N$, and Tr, and positively correlated with IWUE. IXl was strongly positively correlated with IXc, strongly negatively correlated with IC, d, IWHC, WTR, $\Psi$, $\xi_L$, $P_N$, and Tr, and positively correlated with IWUE. IXc was found to be strong negative correlation with IC, d, IWHC, WTR, $\Psi$, $\xi_L$, $P_N$, and Tr, and positively correlated with IWUE. IC was strongly positively correlated with IWHC, WTR, $\Psi$, $\xi_L$, and $P_N$, positive correlation with d and Tr, and negative correlation and IWUE. D was strongly positively correlated with $\Psi$, $\xi_L$, and $P_N$ and positively correlated with IWHC and WTR. IWHC was found to be strongly positively correlated with WTR, $\Psi$, and $\xi_L$, positively correlated with $P_N$ and Tr, and negatively correlated with IWUE. IWUE

was negatively correlated with WTR. WTR was found to be strongly positively correlated with $\Psi$ and $\xi_L$ and showed positive correlation with $P_N$ and Tr. $\Psi$ was strongly positively correlated with $\xi_L$ and $P_N$ and positively correlated with Tr. $\Xi_L$ was strongly positively correlated with $P_N$ and showed positively correlation with Tr. $P_N$ was strongly positively correlated with Tr.

## 4. Discussion

Water balance in plants is essential for plant growth. Leaf water potential can reflect the water status of plants and the influence degree of the soil-vegetation-atmosphere continuous system on the water in plants [39]. It can also reflect the ability of plants to absorb water from soil and maintain plant growth and development [6]. The results showed that leaf water content and water potential of *M. alba* and *B. papyrifera* were parallel to the soil water content. The leaf water content and water potential in waterfront soil were higher than those in arid slope soil, indicating that leaf water content and water potential could respond to soil water deficit immediately. Furthermore, lower leaf water content and water potential indicate soil water deficit, and plants can adapt to water shortage environments by reducing water demand [40].

Photosynthesis is the basis of plant growth and development and one of the indicators to observe the ability of plants adapting to the environmental stress. It is not only affected by the physiological characteristics of plants but also restricted by environmental factors. Water is an important factor involved in photosynthesis. With less soil moisture, photosynthesis will be reduced [41]. Therefore, studying the response of photosynthesis to stress can characterize the adaptability of plants to the environment. [42]. In this study, the $P_N$ and Tr of *M. alba* and *B. papyrifera* in deficient-water soil were lower than those in well-water soil, indicating that soil water deficit would lead to stomatal closure on leaves, weaken transpiration, affect carbon dioxide absorption and reduce net photosynthetic rate, and plants could only maintain their own growth and development with limited water. The results also indicated that photosynthetic indexes could respond to soil water status quickly [43]. At the same time, chloroplasts are the main site of photosynthesis in plants, and low water potential results in damage to chloroplast thylakoid membrane structure, weakened electron transfer and phosphorylation, and decreased photosynthetic rate [44]. The response of photosynthesis to soil moisture was consistent with our previous research results [31].

Changes in electrical signals of plants are usually caused by changes in ion concentration, cell water status and membrane permeability of leaf cells immediately after plants are subjected to environmental stress [33]. In mesophyll cell, the cell and organelle are surrounded by cell membrane, so the cell can be regarded as a concentric circular capacitor with the dual functions of inductance and resistance. The ions and ionic groups in the mesophyll cell are electrolytes [23]. Almost all plant life activities are accompanied by the separation of electric charges, the movement of electrons, the transport of protons and media, etc. [36]. The clamping force stimulates the changes of plant leaves and immediately changes the membrane permeability of leaf cells, resulting in changes in electrolyte concentration and changes in leaf Z, R, Xc, and Xl. According to the Nernst equation, it revealed the Z (or R, Xc, Xl) $= y + ke^{-bF}$ of the theoretical internal relationship between Z, Xc, R, and Xl and the clamping force.

The Nernst equation is used to quantitatively describe the diffusion potential of an ion between A and B systems [25]. This equation relates chemical energy to galvanic electrode potential, making a significant contribution to electrochemistry. Meanwhile, the Nernst equation describes the relationship between the equilibrium potential of an ion and the ion concentration on both sides of the ion-permeable membrane [45]. The results showed that the fitting equations of Z-F, R-F, Xc-F, and Xl-F had good correlation ($R^2 = 0.9620\sim0.9960$, $p < 0.0001$), which highlighted the real existence of the above internal mechanism.

Part of the energy that reduces the internal energy of system converts into external work is defined as Gibbs free energy. In this study, the Gibbs free energy of leaf cell capacitor

is theoretically equal to the work done by the clamping force [25]. The change in Gibbs free energy resulting from a reversible transition from one state to another at constant pressure and temperature is the amount of work available in that state change [46]. According to the Gibbs free energy, it also revealed the theoretically intrinsic relationships between leaves C and the clamping force, and the specific effective thickness (d) was defined. The results showed that the leaf d of *M. alba* and *B. papyrifera* grown in the waterfront soil was significantly higher than those of *M. alba* and *B. papyrifera* grown in the arid slope soil. These results revealed the life phenomenon in plants; that was, when plant water metabolism and growth were vigorous, the electrolyte concentration became low, water was abundant in leaf cell, and leaf d became high.

So far, spontaneous period and intrinsic electrical signals cannot be detected in plants. Most of the electrical signals in plants are induced by a variety of stimuli with low reproducibility [47]. In this study, the intrinsic electrophysiological indices of IZ, IR, IXc, IXl and IC of plant leaves were successfully determined by analyzing the intrinsic relationship between leaf Z, R, Xc and Xl and clamping force. The results showed that IC of leaf in waterfront soil was significantly higher than that in arid slope soil, with lower IZ, IR, IXl, and IXc, but the leaf IZ, IR, IXl, IXc, and IC of *M. alba* grown in waterfront soil and arid slope soil were not significantly different. Drought stress can affect plant water metabolism [5]. The results showed that the IWHC, WTR and water contents of *B. papyrifera* leaves grown under adequate water supply were higher, but there was no significant difference between IWHT and IWUE. In contrast to *B. papyrifera*, the d and IWHC of *M. alba* decreased with decreasing soil moisture, while the IWUE, IWHT and WTR did not change significantly. These results indicated that the intercellular water use characteristics of the two plant species were obviously different and had different response patterns. *M. alba* responded to water shortage by decreasing d. For *B. papyrifera* under water shortage, the effective thickness d of leaves was greatly reduced, the water utilization rate was increased, and the water transfer rate was decreased. These results were mainly due to the fast logarithmic growth period of *M. alba* leaves, short life cycle, low photosynthesis and slow overall growth (d), and water shortage had no effect on water metabolism of plants. However, the leaf life cycle of *B. papyrifera* was long and the growth was fast, which was affected by water deficit at any time. These revealed life phenomena peculiar to plants. In addition, soil water deficit and slowing down of the intracellular water transfer rate resulted in less water demand and therefore sufficient water for plants. As a result, leaf growth would be slow and there would be a larger sacrifice of leaf growth (d). This is consistent with the biological fact that plant biomass affects plant water use efficiency [48].

From Tables 5 and 6, correlation analysis between physiological indices and electrophysiological parameters of the leaves of *M. alba* and *B. papyrifera* was analyzed. As shown in Table 5, electrophysiological and intracellular water information of *M. alba* did not correlate with plant external physiological information (such as $P_N$, Tr, Ψ, and $\xi_L$). However, there were correlations between leaf water potential, water content and photosynthetic parameters, as well as between electrophysiological parameters and intracellular water use parameters, but they were not correlated with IWHT. The results indicated that the leaves of *M. alba* had a short life cycle and leaf function period, and fell off quickly; the leaves were thin, and the inner cells had little water, while the small cells were relatively stable in all aspects under soil water deficit [31]. Plant external physiological information did not affect the intracellular water status, indicating that the intracellular water metabolism of *M. alba* of leaves was independent of environmental water metabolism. Because *M. alba* was an acclimated species for a long time, it had gradually adapted to the shifting environment, and the rapid growth of new leaves was compatible with the activity of continuous leaf picking [3,49]. However, as shown in Table 6, contrary to *M. alba*, the electrophysiological indices of *B. papyrifera* were well correlated, except IWHT. These results showed that the leaves of *B. papyrifera* grew fast, had a long life cycle and leaf function period. The results indicated that both plant intrinsic electrophysiological information and external instant information in *B. papyrifera* were sensitive to soil moisture changes.

*M. alba* and *B. papyrifera* have different adaptive mechanisms to soil water deficit. The intrinsic water use efficiency and water transfer rate of *M. alba* leaves were not affected by the soil water deficit, thus maintaining the constant water retention time in leaves, and the growth of a single leaf was not affected. *M. alba* leaves had a short life cycle, so it responded to environmental changes by reducing the number of leaves. The leaf life cycle of *B. papyrifera* is long, the growth of an individual leaf was slow, and the water transport rate was reduced, in order to maintain the time of water supply to cope with the soil water shortage environment.

Previous studies have shown that the changes of plant electrophysiological parameters Z, R, and C can directly reflect the changes of plant water status [30,50–52]. However, Z, R, Xl, and Xc parameters alone cannot be used to determine water retention, transport and utilization in plant leaf cells [25]. In this study, IWHC, IWUE, IWHT, and WTR were used to accurately reveal the characteristics of intracellular water metabolism in leaves of *M. alba* and *B. papyrifera* in different habitats. The indices in this paper were defined by the intrinsic electrophysiological parameters of plant leaves and had the advantages of stability, accuracy, and representativeness. It is of great significance to accurately describe plant intracellular water metabolism.

## 5. Conclusions

Plant electrophysiological information can rapidly reflect soil water shortage conditions. In this study, the internal relationship between leaf Z, R, Xc, and Xl and the clamping force was revealed and established from the perspective of bioenergetics, and fitting equations were used to calculate the electrophysiological parameters IZ, IR, IXc, IXl, and IC. Then, the leaf intracellular water use parameters d, IWUC, IWUE, IWUT, and WTR of *M. alba* and *B. papyrifera* were defined based on the plant internal electrophysiological parameters. The mechanism of plant water metabolism was evaluated, and the correlation between plant physiological parameters ($\Psi$, $\xi_L$, $P_N$, Tr) and electrophysiological parameters was discussed. The results showed that the intracellular water metabolism based on electrophysiology in *M. alba* and *B. papyrifera* had different response patterns to soil water conditions. *M. alba* gradually adapted to the environment. These conclusions can be used to obtain plant intracellular water information and provide theoretical support for studying the adaptation mechanism of plants to shifting environment.

**Supplementary Materials:** The following are available online at https://www.mdpi.com/article/10.3390/horticulturae8020182/s1, Figure S1: The experimental setup used in the study and a schematic diagram of the parallel-plate capacitor; Formula S1: The deduction process of clamping force and electrophysiological parameter model.

**Author Contributions:** Conceptualization, methodology, funding acquisition, supervision, validation, Y.W.; data curation; formal analysis, writing—original draft, visualization, R.Y.; software, formal analysis, D.X. All authors have read and agreed to the published version of the manuscript.

**Funding:** This research was supported by the project of the National Key Research and Development Program of China (2016YFC0502602), the National Natural Science Foundation of China (No. U1612441-2), the Priority Academic Program Development (PAPD) of Jiangsu Higher Education Institutions, and the Graduate Innovative Projects of Jiangsu Province (2014 (KYLX_1061)).

**Data Availability Statement:** The datasets from (or analyzed during) the current study are available from the corresponding author upon reasonable request.

**Conflicts of Interest:** The authors declare no conflict of interest.

## Abbreviations

C: capacitance, Z: impedance, Xc: capacitive reactance, R: resistance, Xl: inductive reactance, F: clamping force, IC: intrinsic capacitance, IZ: intrinsic impedance, IXc: intrinsic capacitive reactance, IR: intrinsic resistance, IXl: intrinsic inductive reactance, d: specific effective thickness, IWHC: intracellular water-holding capacity, IWUE: intracellular water

use efficiency, IWHT: intracellular water-holding time, WTR: the dynamic water transfer rate. Ψ: water potential, $\xi_L$: leaf water content, $P_N$: the net photosynthetic rate. Tr: the transpiration rate.

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
