# Peer review of "The Differential Response of Intracellular Water Metabolism Derived from Intrinsic Electrophysiological Information in Morus alba L. and Broussonetia papyrifera (L.) Vent. Subjected to Water Shortage"

_horticulturae, doi:10.3390/horticulturae8020182_

Round 1

Reviewer 1 Report

In this paper, the authors presented some interesting results on electrophysiology in Morus alba L. and Broussonetia papyrifera (L.) Vent. under water-limited environment.

Electrophysiology is a precision way to study physiological state of plants.

Although the work is valuable this manuscript is poorly written and too wordy. For example, the introduction should be reduced to 50% because it contains many known and irrelevant information.

The title is not clear. Please mention which specific electrophysiology you are focusing.

I am wondering why these tree species were considered What are their specialties?

In the discussion, the authors need to explain the underlying mechanisms of specific drought responses.

Please concise the results and compare the treatments clearly. Do not use decimal points when comparing values.

I suggest using bar graph rather than the box plot.

Section 3.7 is too long and descriptive.   Please make it concise and easily understandable. No  need to put all values or discuss all values in the results.

Language quality needs improvement.

Author Response

1st Reviewer

Comment 1: In this paper, the authors presented some interesting results on electrophysiology in Morus alba L. and Broussonetia papyrifera (L.) Vent. under water-limited environment.

Response: Thanks very much for reviewer's attention, positive comments and warm work to our manuscript.

Comment 2: Electrophysiology is a precision way to study physiological state of plants.

Response: Thanks very much for the reviewers' positive affirmation of the main research content of our manuscript.

Comment 3: Although the work is valuable this manuscript is poorly written and too wordy. For example, the introduction should be reduced to 50% because it contains many known and irrelevant information.

Response: We sincerely thank the reviewer for the careful reviews. We have simplified it without changing the main content of the article.

Comment 4: The title is not clear. Please mention which specific electrophysiology you are focusing.

Response: We sincerely thank the reviewer for the careful reviews. Plant electrophysiological information can accurately reflect the physiological state of plants. In this paper, a model formula is derived by using the Nernst equation and Gibbs free energy formula characterize the different responses of plant intracellular water metabolism to soil water deficit based on electrophysiology. The parameters impedance (Z) and resistance (R) were determined by LCR device. Then using the formula to calculate capacitive reactance (Xc) and inductive reactance (Xl), and using 3-parameter exponential decay and linear models derive intrinsic impedance (IZ), intrinsic resistance (IR), intrinsic inductive reactance (IXl), capacitive reactance (IXc), capacitance (IC) and specific effective thickness (d). The intracellular water-holding capacity (IWHC), water use efficiency (IWUE), water-holding time (IWHT) and water transfer rate (WTR) of plant leaves on the basis of IZ, IXc, IC and d were defined and applied to reflect intracellular water metabolism. The title has been changed to “The differential response of intracellular water metabolism derived from intrinsic electrophysiological information in Morus alba L. and Broussonetia papyrifera (L.) Vent. subjected to water shortage”. We modified it in the article and marked it in blue. (See line 2-5) 

Comment 5: I am wondering why these tree species were considered What are their specialties?

Response: Thanks very much for the reviewer's careful reviews. Both M. alba and B. papyrifera belong to the Moraceae family. They are woody plants that grow fast and are adaptive. They are commonly grown as medicinal, ornamental and economic plants. At the same time, the adaptability and water use mechanism of these two plant species clearly differed under water deficit condition. In this study, they were used as comparative materials to further study the electrophysiological responses of these two plant species under different water conditions. This is described in the introduction. We marked it in blue. (See line 42-45)

Comment 6: In the discussion, the authors need to explain the underlying mechanisms of specific drought responses.

Response: We sincerely thank the reviewer for the careful reviews. We modified it in the article and marked it in blue. (See line 425-432) 

Comment 7: Please concise the results and compare the treatments clearly. Do not use decimal points when comparing values.

Response: Thanks very much for the reviewer's careful reviews. We modified it in the article and marked it in blue.

Comment 8: I suggest using bar graph rather than the box plot.

Response: We sincerely thank the reviewer for the careful reviews. We modified it in the article. (See Fig.3)

Comment 9: Section 3.7 is too long and descriptive. Please make it concise and easily understandable. No need to put all values or discuss all values in the results.

Response: Thanks very much for the reviewer's careful reviews. I have simplified this part it in the article.

Comment 10: Language quality needs improvement.

Response: We sincerely thank the reviewer for the careful reviews. We modified it in the article and marked it in blue.

Reviewer 2 Report

I have made minor edits to the attached, annotated manuscript.  The use of "clamping" needs to be much better defined. 

Author Response

2st Reviewer

Comment 1: I have made minor edits to the attached, annotated manuscript.  The use of "clamping" needs to be much better defined. 

Response: We sincerely thank the reviewer for the careful reviews. The force exerted on the leaf provided by upper the parallel plate capacitor with different gravity is called the clamping force. The electrophysiological parameters of plant leaves are measured by using the LCR tester. In our previous studies, we have done a lot of experiments on the test parameters and principles. The LCR parallel mode was selected, and the test voltage was optimized to be 1.5V and the frequency was 3kHz. The leaf is first placed between two electrodes of a homemade parallel plate capacitor. Then the data of leaf capacitance, resistance and impedance under different clamping forces are collected continuously. Different clamping forces are achieved by changing the mass of iron. In this study, the clamping force was set up as 1.1N, 2.1N, 3.1N, 4.1N, 5.1N, 6.1N, 7.1N, 8.1N, each iron of the clamping force of 1 N, but as shown in Additional file 1, experiment device of pole and polar wafer is 0.1 N, the electrophysiological parameter changed as the force increased. Sigmaplot curve fitting was used to obtain electrophysiological information.

Comment 2: "...belonging to the family Moraceae."

Response: Thanks very much for the reviewer's careful reviews. We modified it in the article and marked it in blue. (See line 38)

Comment 3: less negative water potential? Please clarify.

Response: We sincerely thank the reviewer for the careful reviews. This sentence is “Researches have reported that the increases of photosynthesis were accompanied by the increased leaf water potential”. We modified it in the article and marked it in blue. (See line 50-51)

Comment 4: "...soil water status...."

Response: Thanks very much for the reviewer's careful reviews. We modified it in the article and marked it in blue. (See line 48)

Comment 5: increases

Response: We sincerely thank the reviewer for the careful reviews. We modified it in the article and marked it in blue. (See line 50)

Comment 6: Misleading. Leaf water potential is only a measure of how tightly water is held at the leaf level. Only when combined with information on time of day collected, previous rainfall or irrigation, and stomatal conductance is it a measure of drought resistance. Please rewrite.

Response: Thanks very much for the reviewer's careful reviews. In the text, we modify it to be “Plant leaf water potential is one of the main physiological indicators reflecting plant soil water status. When combined with stomatal conductance and other information, it can be used as an indicator to measure plant drought resistance”, and marked it in blue. (See line 47-50)

Comment 7: Where did this statement come from?  What clamping force?  From a cuvette when measuring photosynthesis?  Please rewrite.

Response: We sincerely thank the reviewer for the careful reviews. The force exerted on the leaves provided by the upper parallel plate capacitor with different gravity is called the clamping force. When measuring electrophysiological parameters, the leaf was placed between two electrodes of the self-made parallel plate capacitor, and the schematic diagram of the experimental device was shown in Additional file 1. The clamping force is adjusted by adding iron blocks. The more iron blocks are added, the greater the clamping force is. Then the data of capacitance, resistance and impedance values under different clamping forces are continuously collected. The clamping force of the bipolar plate on the experimental device is also the main factor affecting the electrophysiological parameters. This experiment mainly set up 1.1N, 2.1N, 3.1N, 4.1N, 5.1N, 6.1N, 7.1N, 8.1N, then the response curves of electrophysiological parameters to the increasing clamping force were fitted, electrophysiological information could be achieved. The expression in the article has been revised and marked in blue. (See line 70-73)

Comment 8: Who can discuss?

Response: Thanks very much for the reviewer's careful reviews. This is my misrepresentation. We modified it in the article and marked it in blue. (See line 337-338)

Comment 9: "...under adequate water supply..."

Response: We sincerely thank the reviewer for the careful reviews. We modified it in the article and marked it in blue. (See line 390)

Round 2

Reviewer 1 Report

The current version is fine to me.